# Decentralised Voltage Regulation through Optimal Reactive Power Flow in Distribution Networks with Dispersed Generation

Edoardo Daccò *, Davide Falabretti, Valentin Ilea, Marco Merlo, Riccardo Nebuloni and Matteo Spiller

Department of Energy, Politecnico di Milano, 20156 Milano, Italy; davide.falabretti@polimi.it (D.F.); valentin.ilea@polimi.it (V.I.); marco.merlo@polimi.it (M.M.); riccardo.nebuloni@polimi.it (R.N.); matteo.spiller@polimi.it (M.S.)
* Correspondence: edoardo.dacco@polimi.it; Tel.: +39-02-2399-8505

**Abstract:** The global capacity for renewable electricity generation has surged, with distributed photovoltaic generation being the primary driver. The increasing penetration of non-programmable renewable Distributed Energy Resources (DERs) presents challenges for properly managing distribution networks, requiring advanced voltage regulation techniques. This paper proposes an innovative decentralised voltage strategy that considers DERs, particularly inverter-based ones, as autonomous regulators in compliance with the state-of-the-art European technical standards and grid codes. The proposed method uses an optimal reactive power flow that minimises voltage deviations along all the medium voltage nodes; to check the algorithm's performance, it has been applied to a small-scale test network and on a real Italian medium-voltage distribution network, and compared with a fully centralised ORPF. The results show that the proposed decentralised autonomous strategy effectively improves voltage profiles in both case studies, reducing voltage deviation by a few percentage points; these results are further confirmed through an analysis conducted over several days to observe how seasons affect the results.

**Keywords:** voltage regulation; renewable distributed energy resources; distribution network; optimal reactive power flow; local control laws





## 1. Introduction

In recent years, the world's capacity to generate electricity from renewable resources has expanded faster than ever [1], creating a real chance of tripling global capacity by 2030 [2–4]. In 2023, the new renewable energy capacity added to the world's energy systems grew by 50%, reaching almost 510 GW, with solar photovoltaic (PV) accounting for three-quarters of additions. In Europe, the growth rate of renewable capacity is expected to double in 2023–2028 compared with the previous six years, with an additional 532 GW [1]; in particular, distributed solar PV will continue to be the main source of expansion, supported by new feed-in tariffs, tax exemptions, innovative automation functions, and improved battery performance to mitigate the uncertainty of PV production [5].

In this new scenario, characterised by the massive penetration of renewable and non-programmable Distributed Energy Resources (DERs), distribution system operators (DSOs) have to face an increasingly complicated set of challenges [6]: indeed, given their intermittent nature and dependence on the variability of weather conditions, DERs may cause negative impacts on the power grid, including an increase in reactive power flows [7], voltage fluctuations [8–10], unbalances [11,12], and congestions [13].

Voltage quality represents one of the aspects of distribution networks (DNs) operation that is more affected by the presence of a large number of DERs [14]; DSOs must provide a reliable and stable power supply voltage for the proper functioning of all electrical appliances, allow a reliable operation of generating units [15], and prevent damage to the

network infrastructure. For this reason, voltage regulation plays a key role in ensuring that the voltage profiles remain within acceptable limits: according to the technical standard EN 50160 [16], the supply voltage on MV and LV DNs in normal conditions needs to be within ±10% with respect to the reference voltage level.

Traditionally, the on-load tap changer (OLTC) of high-voltage/medium-voltage transformers has long been utilised in primary substations (PS) to keep the voltage close to the nominal value by compensating for generation and load pattern changes and voltage variation on the transmission grid. However, due to an increased fluctuation in the voltage profiles over the distribution lines caused by load and DER generation variations, the regulation performed through the OLTC in PS alone may be insufficient to ensure that the voltage value over the entire DN remains within the admitted thresholds [17]. For this reason, the introduction of decentralised voltage control is often identified as a possible solution to improve the voltage quality over modern DNs.

In this framework, DERs, especially inverter-based, are pivotal because they can regulate their reactive power (and, in exceptional conditions, active power) to maintain the voltage at the desired level. For this purpose, technical connection rules for DERs in many countries have been updated to include a minimum set of reactive (and active) power control functions to support the network during contingencies. In Italy, technical standards CEI 0-16 [18] and 0-21 [19] provide a set of prescriptions concerning DER control capabilities, harmonised with the relevant ENTSO-E network code [20]. DERs can use these newly added control laws to act as distributed voltage regulators, improving the voltage at their terminals and reducing the occurrence of over/under-voltage events on the grid. This way, DERs can provide low-cost and fast-timescale reactive power compensation throughout the DN, reducing the mechanical switching burden on traditional devices and improving voltage profiles even in the presence of high penetration of intermittent generation.

In the present framework, this paper aims to lay the foundations for developing a voltage control method on DNs through a decentralised strategy that copes with the main limitations of the current voltage strategies. In particular, at present, the voltage value at the DERs' terminals is usually controlled according to local voltage control laws, without any communication among the various DERs. Often, the voltage control exploits a droop function that regulates the reactive setpoint of each DER as a function of the voltage locally measured at the point of connection with the grid. Usually, the droop coefficient of the voltage control function (i.e., its angular coefficient) is the same for all the DERs involved. Despite its simplicity, this technique is not optimal, since the voltage sensitivity over the reactive power changes as a function of the grid's topology and operating conditions [7]. To overcome this issue, this work proposes a customised optimal reactive power flow (ORPF) that allows a minimising of the deviation between the voltage value in each node and its nominal value (by optimally setting up the droop coefficient of each DER), while exploiting the specific control capabilities of each generator. To this end, the work provides a procedure to optimally set up the DER droop control, exploiting functionalities already defined in network codes. The proposed methodology is tested on both a small-scale test grid and a real Italian distribution grid to propose an efficient solution to the voltage regulation problem.

This work is structured as follows. Section 2 provides an extensive overview of the relevant literature, including the key theoretical frameworks and relevant studies. Section 3 describes the methodology, highlighting the distributed control law and the mathematical model that was implemented. Section 4 provides an illustrative small-scale case study to highlight the potential and peculiarities of the proposed decentralised control. Furthermore, this section evaluates the case study and the results obtained from the numerical simulations performed on the real existing distribution grid. Finally, Section 5 gives some conclusions based on the study's outcomes.

## 2. Literature Review

In this section, a deep bibliographical review is performed. In Section 2.1, a careful analysis is carried out concerning the various voltage regulation strategies; in Section 2.2, a focus is provided on ORPFs as one of the methodologies today most commonly used to cope with the voltage regulation problem.

### 2.1. Voltage Regulation in Modern Distribution Networks

The heightened focus on sustainable and renewable energy systems has triggered a significant transition in modern DNs; due to the growing demand for cleaner and more efficient energy solutions, understanding the dynamics of DERs within current DNs becomes imperative [21]. In this regard, it is crucial to note that the implementation of such resources into the electrical network is not without challenges [22]: in particular, the presence of DERs could lead to a voltage rise, depending on the active and reactive power exchanged, as well as the network's topology. The DER connection to a distribution feeder leads to a voltage rise at the generator's delivery point, as explained in [22–24]. Furthermore, power imbalances between DER production and load demand tend to create over/under-voltage events to some extent.

Traditionally, to keep the voltage within limits mandated by the regulations, DSOs could increase the conductor size [25], install voltage regulators [26], or change the setpoint of primary or secondary transformer taps [27]. However, the fast growth of DERs in DNs requires fast structural interventions that can hardly keep up with the pace of the energy transition [28]. The best solution to cope with the rhythm of the energy transition resides in adequate control strategies of distributed generators. In this regard, DSOs developed various voltage controls to improve voltage quality over MV/LV grids: centralised control, decentralised autonomous control (or local), and decentralised coordinated control [29–31]. DSOs, which implement centralised control, send command signals to DERs [32], OLTCs, or STATCOMs [33]; they act as central coordinators and network supervisors in the centralised mechanism, communicating with all the network agents to fully exploit the resources' flexibility. However, centralised control requires the coordinator to constantly have access to all the information needed to operate the grid, which is not always feasible or economically viable. Hence, the availability and reliability of communication links among voltage sensors, voltage control devices, and the central entity's control centre constrain the system's efficiency [34].

On the other hand, decentralised control presents greater flexibility and scalability [35]. The decentralised coordinated voltage strategy relies on communication among DERs to update their setpoint through local computations. The system exploits a large amount of data to identify a robust solution for the voltage regulation problem. Due to the repeated data exchange and the reduced deployment in actual application, distributed coordination does not seem to be a ready-to-use technology for voltage regulation in distribution systems [36]. Regarding decentralised voltage strategies, the literature widely exploits multi-agent schemes to enhance voltage regulation. Paper [37] proposes a new voltage control scheme to cope with DER penetration; the authors exploited a 6.6 kV test grid as a benchmark. Paper [38] proposes an innovative multi-agent graph-based deep reinforcement learning tested on the IEEE 33-bus and 123-bus distribution test feeders. Paper [39] studies a novel physics-informed multi-agent deep reinforcement learning voltage control methods tested on the IEEE 33-bus and IEEE 141-bus systems. Despite the versatility of multi-agent schemes, they always require big data handling and a huge computational effort, and their convergence is challenging [40,41]. Furthermore, the studies proposed in the literature focused on punctual action by specific DERs; moreover, the DSO could find some approaches that generalise the voltage control for system stability purposes more appealing.

*2.2. Optimal Reactive Power Flow*

ORPFs are complex, non-linear, and non-convex problems, essential mathematical programming for studying power grids' proper operation and control [42]. The scope of an ORPF is to minimise the overall cost of the system by adequately dispatching the optimal reactive power setpoint of each production unit. Moreover, in DNs, an ORPF could also manage PS' OLTCs and capacitor banks [43]. The management of these elements by optimisation allows for the identification of their setpoints to operate the grid safely and efficiently. Numerous objective functions have been investigated in the literature, such as the kVAR cost, total fuel cost, active power transmission losses, and voltage deviation/stability [44,45]. It is worth noticing that different objective functions return different optimised outcomes, and their adoption could be case-specific.

Mathematical programming exploits algorithms to identify the value of the variables that ensure the global optimal of a given objective function. The major convergence problem in mathematical programming resides in binary variables. This element inside an optimisation divides the problem's feasibility area, limiting the solver's efficiency. Moreover, the power flow equations are non-linear and non-convex due to the presence of trigonometric components, which may lead to local optimal solutions instead of global ones. The linearisation of power flow equations is discussed, allowing the linearisation of trigonometric expression [46]. The approximation of the power flow equation leads to an error proportional to the angle considered in the trigonometric function [47]. The voltage angle has a low value in the DN because of the limited electrical distance of the lines; on average, the linearisation error has a value of around 1% for the DN [47]. It is also important to note that the implementation on DNs of standard ORPF methods is not straightforward, due to a lack of proper monitoring devices [48] and the acquisition of real-time measurements; consequently, alternative methods capable of managing the problem more effectively are usually preferable [49–51].

In this regard, this work proposes a partially autonomous decentralised control strategy in which DER units autonomously operate in response to voltage deviations at their terminals; generally, DERs in DNs embed a voltage droop control to adjust their reactive power to compensate for measured voltage deviations. With this strategy, the data exchange among the different resources involved in the regulation is limited, because the DSO sends setpoint signals to the dispersed units with a predefined (low) periodicity (e.g., daily or monthly). Then, in real time, each DER on the grid autonomously regulates the voltage at its terminal through a local voltage droop characteristic. A customised ORPF algorithm provides efficient angular coefficent setpoints to each DER; the generalisation of this information allows the creation of a rule or strategy for DSOs to manage the electric power system more efficiently.

## 3. Methodology

This work proposes an ORPF-based decentralised voltage regulation. The idea behind this method is to control the angular coefficient of a linear local voltage control law of each DER unit over the DN network: this parameter (voltage droop) adjusts the reactive power contribution of each distributed resource, according to the voltage measured at its terminals (Q(V) control law). The ORPF allows for the identification of the optimal angular coefficient that minimises voltage deviations from the nominal values. It should be noted that the optimisation based on the ORPF mathematical model does not directly modify the reactive power contribution of each DER but acts on the droop of the voltage control law: this aims to avoid constant communication between the DSO and DERs, improving the efficiency and robustness of the proposed strategy. Indeed, a communication signal is sent by the DSO to the dispersed units with a predefined periodicity (e.g., daily or monthly) to set the optimal angular coefficient of the voltage control, leaving DERs to autonomously adjust the reactive power at their terminals according to the control law.

In this section, the proposed methodology is explained in detail; in Section 3.1, the droop voltage control law is defined according to the Italian CEI 0–16 technical standards. In Section 3.2, the numerical model of the ORPF is described, and the characterising equations are provided for all the reported models.

### 3.1. The Q(V) Control Law

In the present study, to locally control the voltage profile at the DER terminals, the Q(V) control law reported in Figure 1 has been implemented: it modifies the reactive power exchanged by the DER unit ($Q$), both in absorption and injection, based on the voltage ($V$) measured at the generator point of connection. Even for small fluctuations around the reference voltage value (1.0 p.u.), the regulation is supposed to adjust the reactive power (no deadband regulation); instead, if the voltage value is outside the range $V_i \div V_s$, the regulation saturates at $\pm Q_{max}$. According to technical standards, $Q_{max}$ is defined as 0.4843 of the DER nominal active power ($P_{max}$), corresponding to a power factor of 0.9. The equation that describes this control law is reported next:

$$
Q = \begin{cases}
Q_{max} & if \ V < V_i \\
m \cdot P_{max} \cdot \left( V - V_{ref} \right) & if \ V_i \leq V \leq V_s \\
-Q_{max} & if \ V > V_s
\end{cases}
\tag{1}
$$

In this case, $V_{ref}$ has been set equal to 1.0 p.u., while $m$ is the negative angular coefficient that controls the reactive power exchange by the DER as a function of the measured voltage value; the developed novel ORPF algorithm optimises this parameter.

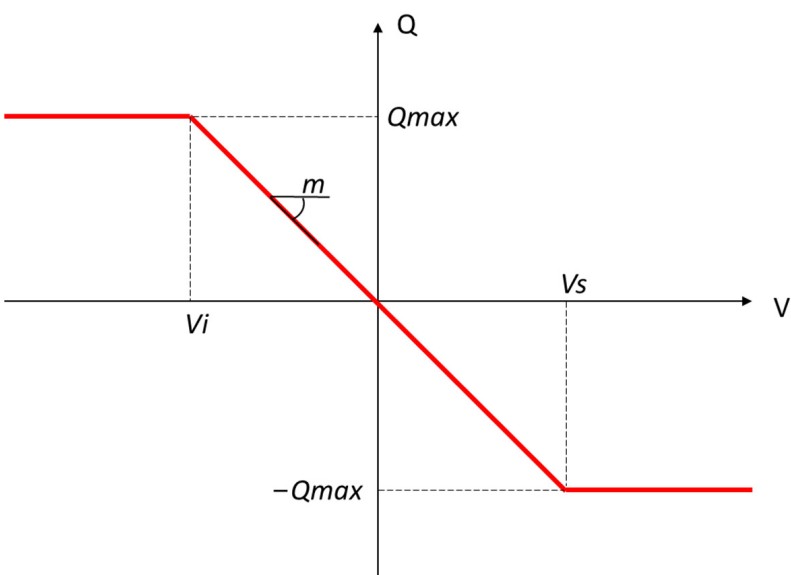

**Figure 1.** Schematic representation of the Q(V) control law.

With the proposed approach, the Q(V) control law is modified, as shown in Figure 2. In particular, the angular coefficient $m$ is varied to obtain different reactive power contributions for the same voltage deviation; therefore, $\pm Q_{max}$ is reached for different voltage levels of $V_i$ and $V_s$. The angular coefficient $m$ and the voltage thresholds will depend on the output of the optimisation process described in the next subsection. It is defined to obtain the maximum benefit from the regulation capabilities of each DER. It is worth noticing that the voltage at each system node cannot overcome $V_{min}$ and $V_{max}$, respectively, equal to 0.90 and 1.10 p.u.

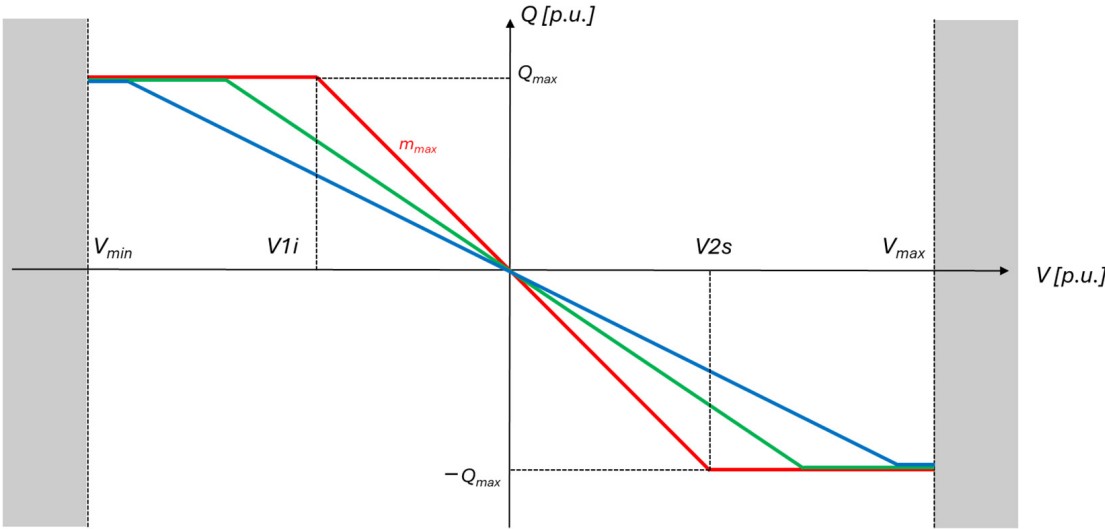

**Figure 2.** Different Q(V) characteristics.

### 3.2. The Mathematical Model

The ORPF problem is an optimisation problem which consists of the minimisation of an objective function subject to a set of constraints that express the correct operation of the system investigated. In turn, constraints are subdivided into equality constraints, such as power flow equations, and inequality constraints, such as limits on voltages, currents, generator capabilities, etc.

Regarding equality constraints, the set of power flow equations that physically describe the point of operation of the network can be defined as:

$$P_g(k) - P_d(k) = \sum_{m \in \mathbb{B}} V_k V_m Y_{km} \cos(\delta_k - \delta_m - \theta_{km}), \ \forall k \in \mathbb{B} \tag{2}$$

$$Q_g(k) - Q_d(k) = \sum_{m \in \mathbb{B}} V_k V_m Y_{km} \sin(\delta_k - \delta_m - \theta_{km}), \ \forall k \in \mathbb{B} \tag{3}$$

where:

- $\mathbb{B}$ is the set of all nodes of the grid;
- $P_g(k)$ is the active power injected in the $b^{th}$ bus;
- $P_d(k)$ is the active power required by the $b^{th}$ bus;
- $Q_g(k)$ is the reactive power injected in the $b^{th}$ bus;
- $Q_d(k)$ is the reactive power required by the $b^{th}$ node;
- $V_k$, $V_m$ are the magnitudes of the nodal voltage in p.u.;
- $\delta_k$, $\delta_m$ are the angles of the nodal voltage in p.u.;
- $Y_{km}$ is the magnitude of the $km^{th}$ element of the nodal admittance matrix;
- $\theta_{km}$ is the angle of the $km^{th}$ element of the nodal admittance matrix.

Regarding inequality constraints, the nodal voltages of the network and the thermal limits of the lines are constrained by the technical minimum and maximum limits:

$$V_k^{min} \leq V_k \leq V_k^{MAX}, \ \forall k \in \mathbb{B} \tag{4}$$

$$I_{km} \leq I_{km}^{MAX}, \ \forall k, m \in \mathbb{B} \tag{5}$$

As already introduced, the objective function is the voltage deviation minimisation: the model tries to modify the generator setpoint of $m$ to set the nodal voltages as close as possible to the nominal value by DER reactive power flows. Although different objective functions have been investigated in the literature [52], the voltage deviation minimisation has been shown to be the main goal of DSOs. Even if other factors could be considered in the

optimisation (e.g., losses), their relevance from the DSOs' perspective is often questionable and it widely depends on the quality of service standards and regulatory prescriptions in place. In addition to voltage deviations, another contribution has been added to the objective function that aims to limit the value of the angular coefficient *m* of the Q(V) control law, avoiding any steep change in the voltage droop characteristic; steep changes should be avoided because the droop control would saturate to $\pm Q_{max}$ even for small voltage errors. Consequently, the active power production would be reduced to keep the operating point within the DER capability curve, thus creating economic problems for the plant operator.

To this end, the implemented objective function is:

$$OF = \min\left(\sum_{k \in \mathbb{B}} |V_k - V_k^n| - \sum_{g \in \mathbb{G}} C \cdot m_g\right) \qquad (6)$$

where:

- $\mathbb{G}$ is the set of nodes of the grid at which the DERs are connected;
- $C$ is a suitable penalty factor;
- $m_g$ is the angular coefficient of the $g^{th}$ distributed generator.

As already introduced, it is worth noticing that the optimisation function does not directly control the reactive power setpoint of each DER but the angular coefficient of the Q(V) control law. This approach is designed so that, at the beginning of each considered period (e.g., one day), the DSO runs the ORPF algorithm to identify the optimum *m* based on a dataset of historical measurements (e.g., the measurements collected on the same day of the previous week). The ORPF identifies the *m* as the most efficient values that minimise the voltage deviation. The optimal angular coefficients are sent through a suitable communication channel to the DERs, which implement the required regulation autonomously over the period under analysis. The structure of the presented decentralised voltage regulation is shown in Figure 3. In particular, the *m* signals are sent to each DER on a daily basis (*D1*, *D2*, *D3*), at the beginning of the day (*t* = 0); for the rest of the day, DERs regulate autonomously, according to the received angular coefficient setpoint.

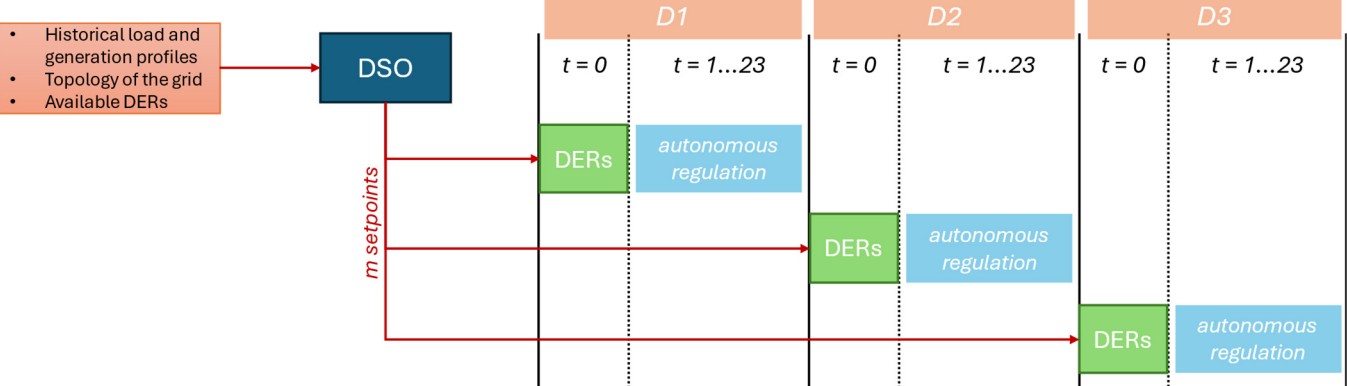

**Figure 3.** Schematic representation of the presented decentralised voltage regulation.

## 4. Case Studies and Numerical Results

This section aims to prove the effectiveness of the proposed ORPF algorithm on different case studies. Section 4.1 presents its application to a small-scale DN [53], while in Section 4.2, the proposed methodology is applied to a real-life DN in Italy. All the significant results are reported in the following. The NLP optimisation problem has been implemented in GAMS 38.3.0 [54] and solved using KNITRO 13.0.0 [55].

### 4.1. Test on a Small-Scale Distribution Network

The automatic decentralised voltage control algorithm has been tested on a small-scale DN (Figure 4) to check the effectiveness of the proposed method. The network consists of an MV busbar, an MV line, an MV load, and a distributed generator; a short line connects

the DER to the load busbar. The MV busbar represents the slack bus of the system, with the voltage magnitude fixed at 1.0 p.u. and the angle at $0°$.

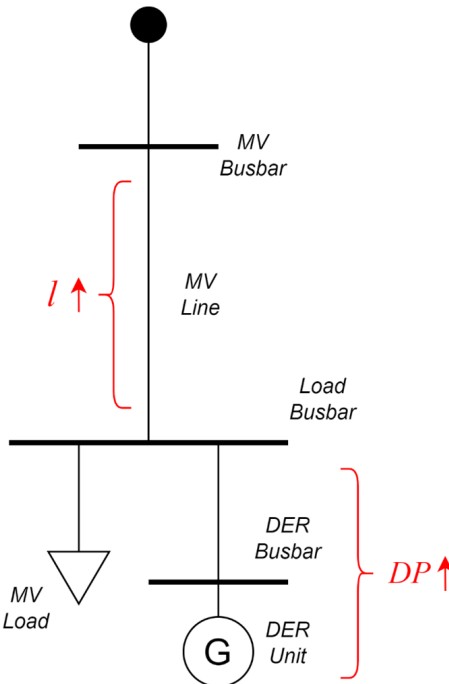

**Figure 4.** Schematic representation of the small-scale distribution network.

To evaluate the performance of the optimisation algorithm, two parameters of the test grid are changed iteratively; these parameters are the length of the MV line $l$, and the ratio $DP$ between the active power produced by the DER and the one absorbed by the load, obtained by iteratively increasing the power produced by the DER according to the formula in Table 1. In more detail, Table 1 lists the parameters investigated in the test grid, for which the parameters with apex "0" refer to the initial value. The line parameters are taken from the datasheet of Prysmian [53], and the 7 MW load power had been selected as the nominal capacity of an MV (15 kV) line.

**Table 1.** Variations of production/load active power ratio and length of the small-scale distribution grid line.

| Production/Load Active Power Ratio $DP$ | Line Length $l$ |
| :---: | :---: |
| $\begin{cases} P_{load} = 7 \text{ MW} \\ cos\varphi_{load} = 0.95 - Inductive \\ P_g = P_g^0 + 0.5 \cdot (h-1) \\ P_g^0 = 0 \text{ MW} \end{cases}$ | $l = l^0 \cdot k$ <br> $l^0 = 0.217$ km <br> $\begin{cases} R^0 = 0.042 \ \Omega \\ X^0 = 0.020 \ \Omega \\ B^0 = 13.480 \ \mu S \end{cases}$ |
| $h = 1:41$, generator growth index <br> $k = 1:73$, line-length growth index | |

The following subsection investigates how the variation of these parameters affects the Q(V) control law identified by the optimisation algorithm.

Numerical Results: Minimisation of Voltage Variations

In this subsection, the effects of some electrical parameters of the test network on the voltage regulation via the Q(V) control law are evaluated; in particular, it is investigated how the line length $l$ and the ratio $DP$ between generation and load impact the value of the angular coefficient $m$ of the Q(V) control law. In addition, the penalty factor $C$ of the multi-

objective function (7) is varied to study the sensitivity of the angular coefficient *m* with respect to the voltage deviation term. Three different penalty factors are considered, namely, $C = 0$, $C = 10^{-7}$, and $C = 10^{-5}$. In particular, the first penalty factor has been chosen to study the proposed decentralised voltage regulation with an objective function that only takes the voltage variation into account and does not restrict the angular coefficient of Q(V); the other two penalty factors have been defined according to a sensitivity analysis: as a result of the tuning process, these values proved to be the most efficient trade-off between the penalty on the droop coefficient and the minimisation of the voltage deviation.

In the first instance, the ORPF strategy is evaluated, and the objective function is set to contain only the voltage deviation term, i.e., the penalty factor *C* on the angular coefficient of the Q(V) control law is equal to zero. Therefore, *m* can take any value to keep the voltage as close as possible to the reference value. In Figures 5 and 6, the relationship between the angular coefficient *m* of the Q(V) control law has been plotted as a function of the length of the line *l* and the active power ratio *DP*, respectively. It is worth noticing that, in both figures, there are no clear-cut trends for the value of the angular coefficient *m*, either with respect to the line length or the DER penetration into the test network. Furthermore, the *m* parameter assumes very high values, reaching more than 1000 p.u.; thus, the Q(V) control law is a stepped regulation, which saturates at $\pm Q_{max}$ for very low voltage deviations. This regulation could be detrimental to the network operation, bringing instabilities in the voltage control. Consequently, the null penalty factor in the ORPF algorithm is not recommended and discarded in the following analysis.

Next, a penalty factor has been inserted into the objective function to drive the outcome of the ORPF towards a realistic trend. A $10^{-7}$ penalty factor has been chosen to have a small, although not negligible, contribution of the *C* parameter on the objective function; in this case, the ORPF algorithm has an objective function that considers both the voltage deviation and the angular coefficient *m* of the Q(V) control law. In Figures 7 and 8, the relationship between the angular coefficient *m* of the Q(V) control law has been plotted with respect to the length of the line *l* and the active power ratio *DP*. It should be noted that, in this case, the values of *m* are smaller than in the previous case, and, in both figures, specific trends are outlined.

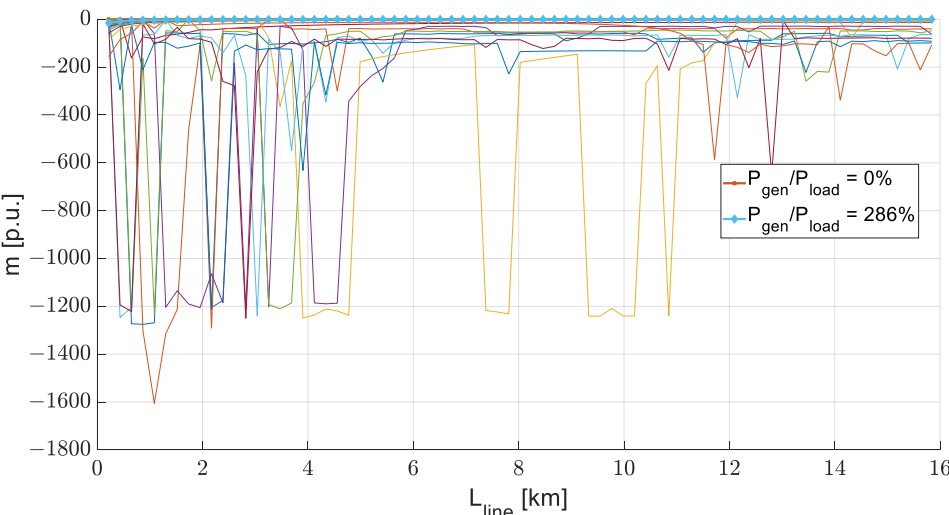

**Figure 5.** Variation of the angular coefficient of the Q(V) characteristic with respect to the length of the line for C = 0.

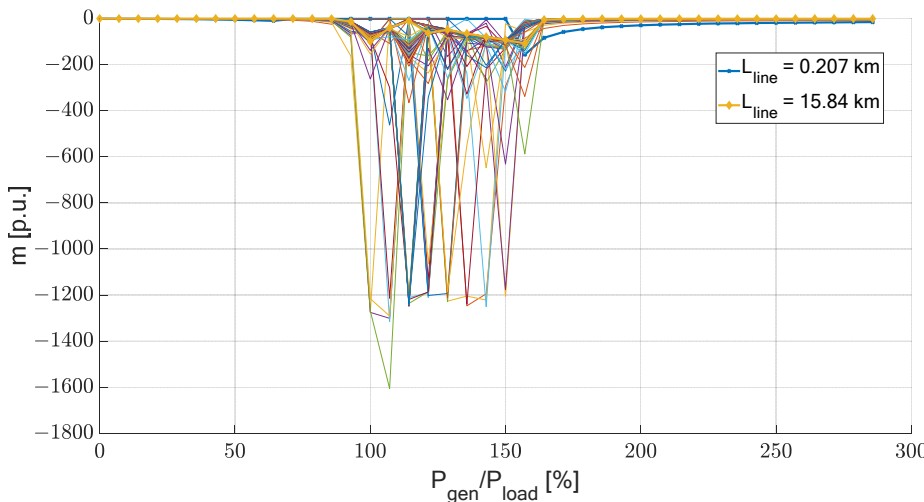

**Figure 6.** Variation of the angular coefficient of the Q(V) characteristic with respect to the active power ratio for C = 0.

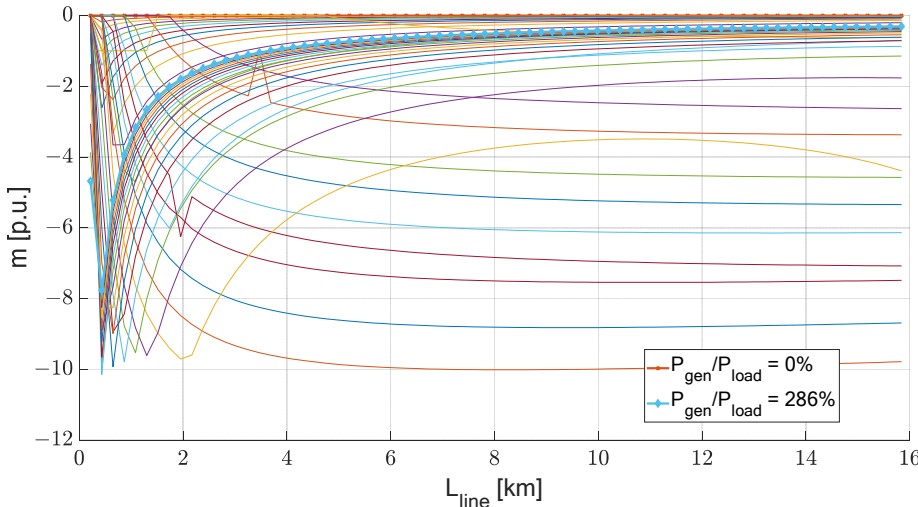

**Figure 7.** Variation of the angular coefficient of the Q(V) characteristic with respect to the length of the line for C = $10^{-7}$.

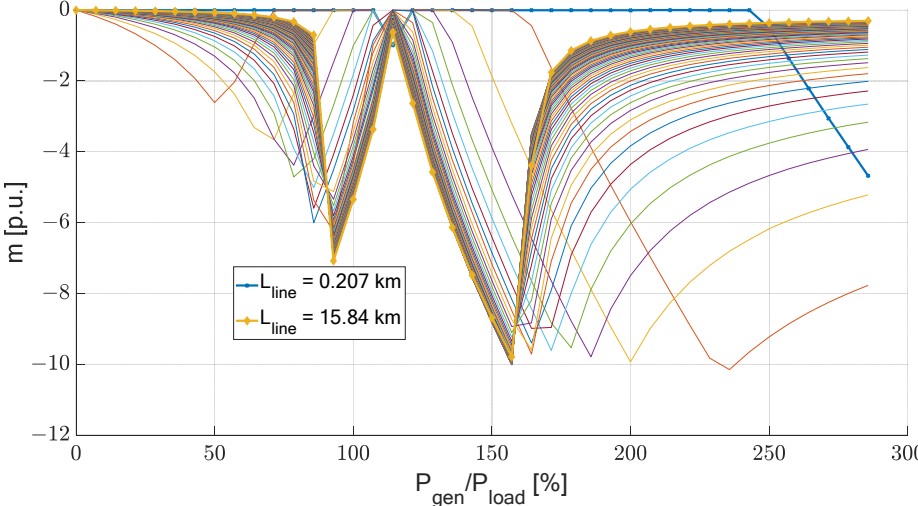

**Figure 8.** Variation of the angular coefficient of the Q(V) characteristic with respect to the active power ratio for C = $10^{-7}$.

For a better understanding of the obtained trends, Figures 9 and 10 report three specific case studies relevant to the distributed voltage regulation performed. In particular, the minimum and maximum cases were chosen for the length of the line *l* and the ratio *DP*; an intermediate case was also selected: *DP* = 100% and *l* = 9 km.

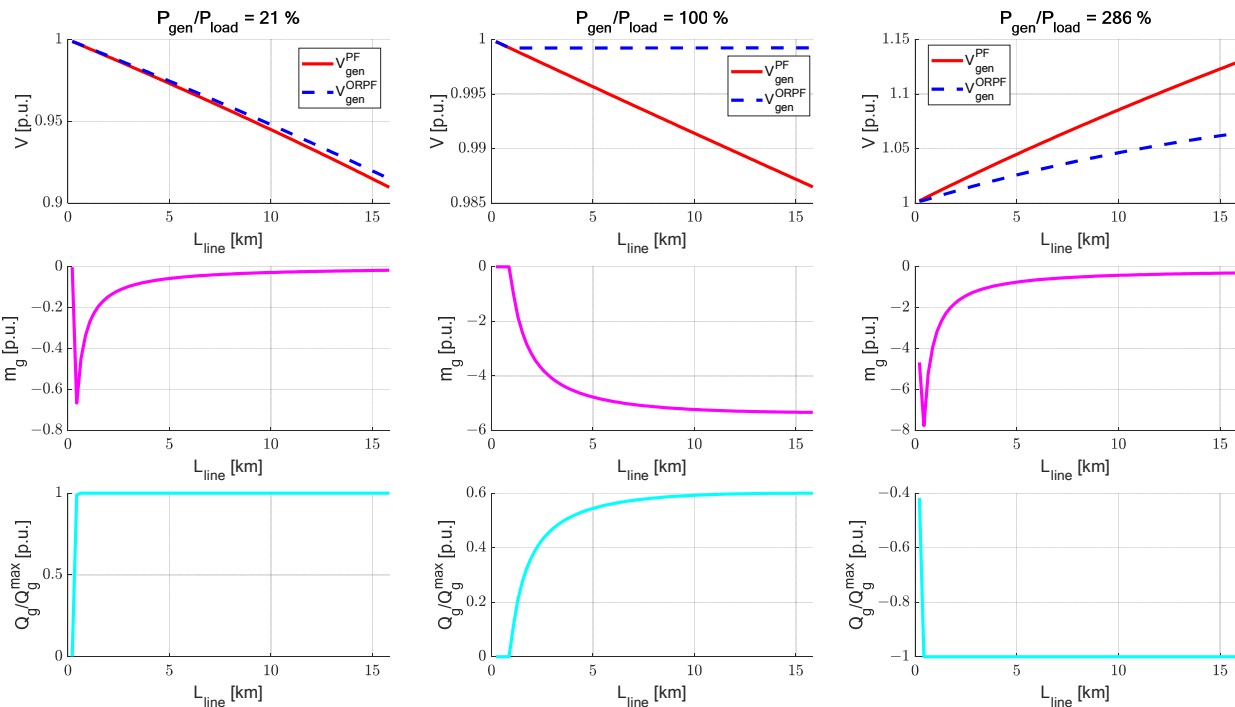

**Figure 9.** Variation of the voltage, angular coefficient of the Q(V) characteristic, and reactive power produced with the line length for C = $10^{-7}$.

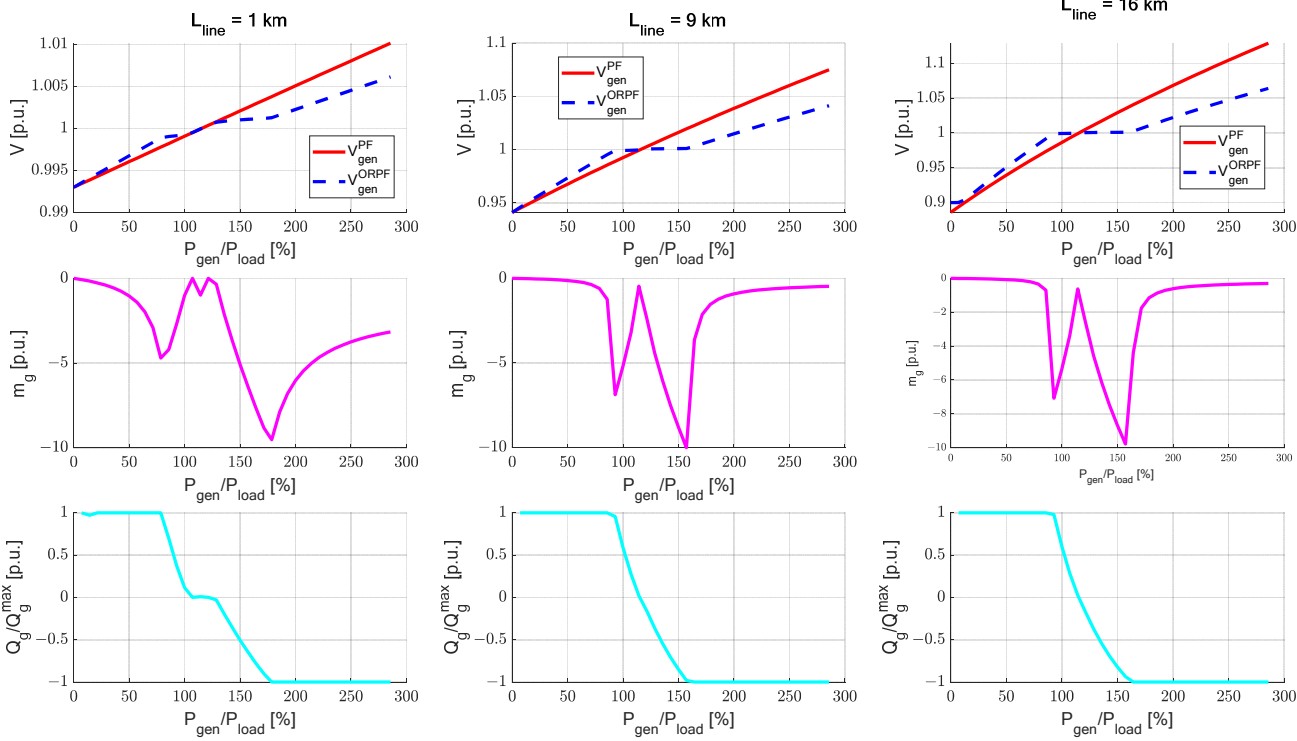

**Figure 10.** Variation of the voltage, angular coefficient of the Q(V) characteristic, and reactive power with active power ratio for C = $10^{-7}$.

Looking at the first figure, the ORPF strategy has a beneficial effect on the voltage. In all three cases, the regulated voltage $V_{gen}{}^{ORPF}$ is closer to the reference value than in the voltage without regulation $V_{gen}{}^{PF}$. This is due to the voltage regulation carried out through the droop control law; indeed, the reactive power exchanged by the DER saturates at the maximum permissible value ($\pm Q_{max} - 1.0$ p.u.), showing that the distributed generation regulates the voltage at its terminals. This is also confirmed by the angular coefficient curves: as the line length increases, the voltage increases, exceeding the reference value, and the reactive power delivered by the DER saturates. As the length increases, the voltage keeps increasing, moving the saturation point of the Q(V) characteristic further from the reference value. For this reason, the angular coefficient $m$ is reduced, while still providing the maximum reactive power.

Similar reasonings can also be applied with respect to power variations. In this case, the angular coefficient of the Q(V) control law is strongly influenced by the voltage at the generator's terminals: for *DP* values between 100% and 150%, the voltage is close to the reference value. For this reason, the voltage regulation is not activated, dropping the DER reactive power to zero; thus, the angular coefficient also goes to zero. For all other *DP* values, it is observed that the regulated voltage $V_{gen}{}^{ORPF}$ is better than the voltage without regulation $V_{gen}{}^{PF}$, via the activation of the Q(V) control law, which identifies the best angular coefficient that saturates at $\pm Q_{max}$ the reactive power value.

For a complete understanding, the case where the penalty factor $C$ is set to $10^{-5}$ was also investigated. In Figures 11 and 12, the same cases as previously depicted have been represented.

The next figure (Figure 13) reports the occurrences of voltage variations from the reference value. It is observed that the larger the penalty factor $C$, the greater the voltage deviation; however, by increasing $C$, the worsening is not significant; thus, it is still preferable to choose penalty factors other than zero to limit the value of $m$, while still providing reactive power by the DERs. In the next subsection, $10^{-5}$ has been chosen as the penalty factor for a real DN case study.

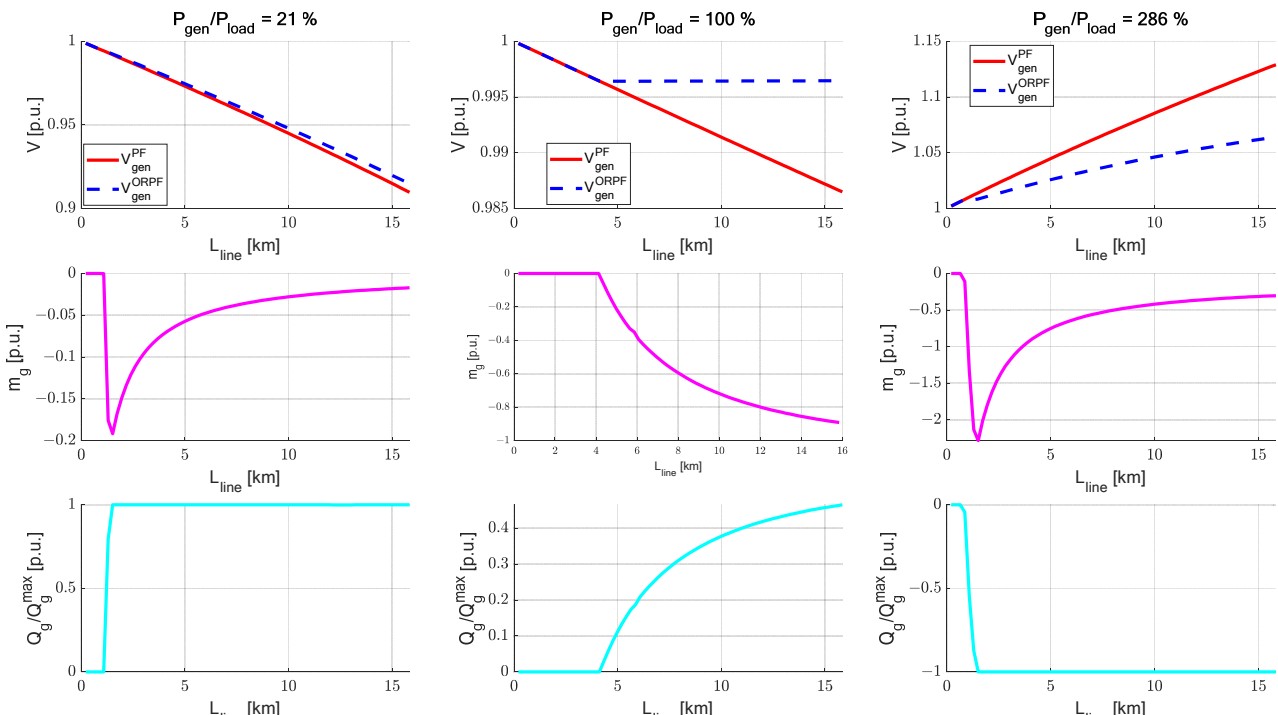

**Figure 11.** Variation of the voltage, angular coefficient of the Q(V) characteristic, and reactive power with line length for C = $10^{-5}$.

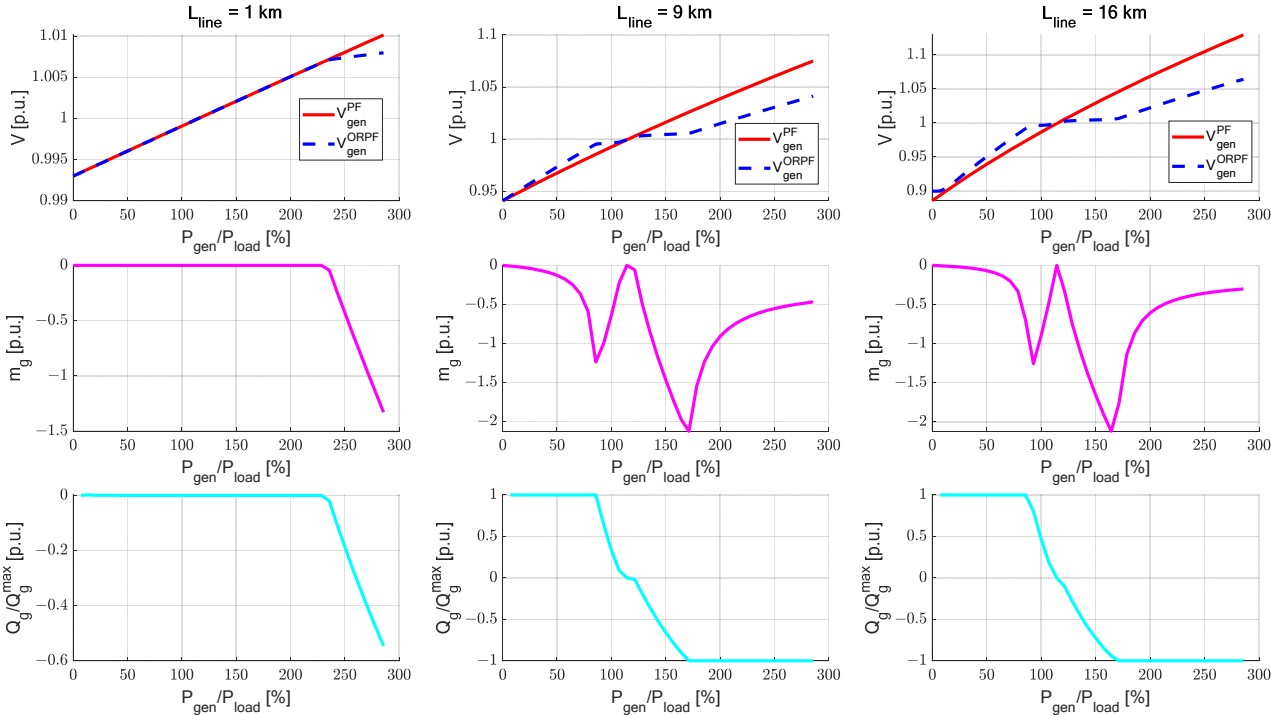

**Figure 12.** Variation of the voltage, angular coefficient of the Q(V) characteristic, and reactive power with the active power ratio for C = $10^{-5}$.

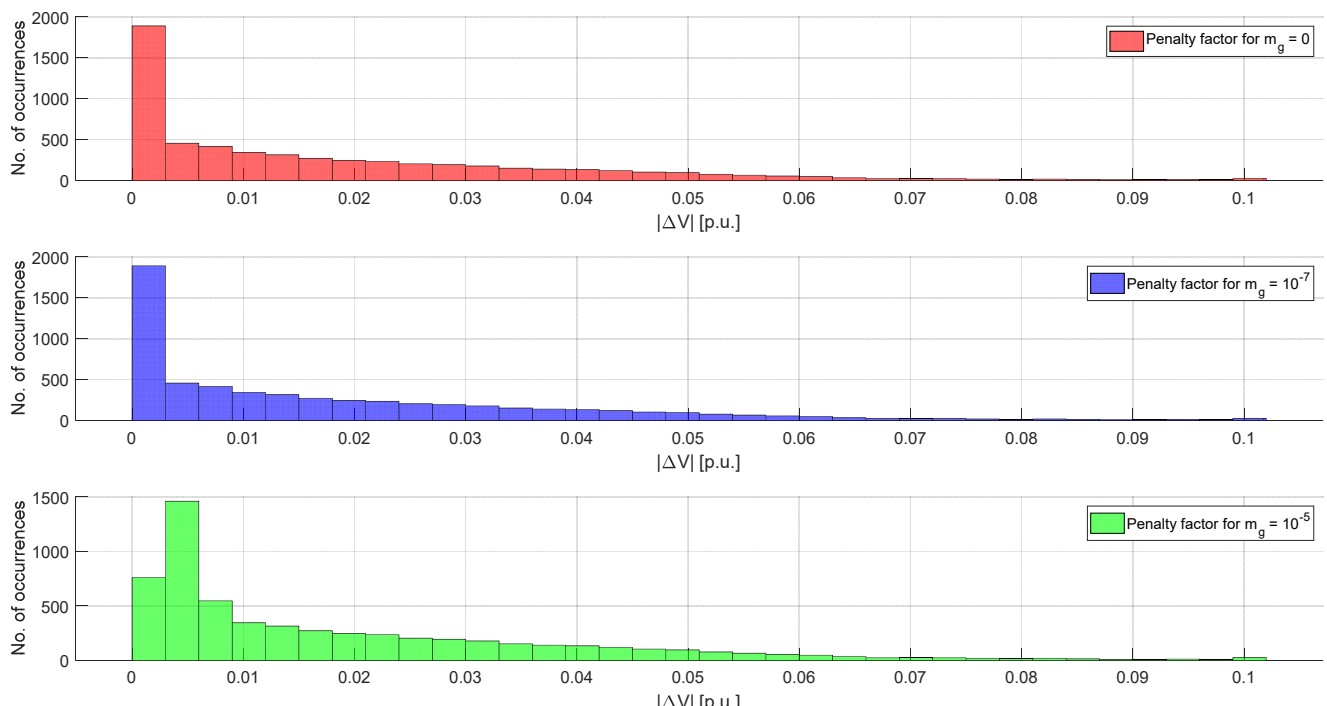

**Figure 13.** Number of occurrences of voltage deviations (absolute value) with respect to the penalty factor.

### 4.2. The Aosta Case Study

The proposed approach's effectiveness has also been evaluated in a case study relevant to the distribution grid that covers Aosta, a medium-sized city in the northwestern part of Italy. A PS, in which two 31.25 MVA transformers are present, feeds 17 feeders with an average load of 2 MW and a 16 MW peak power. Each feeder is accurately described by

data about resistance, inductance, and capacitance; furthermore, the Aosta grid has almost 200 secondary substations. A more detailed grid description can be found in [56].

Figure 14 provides a schematic representation of the PS; in particular, only the MV portion of the distribution grid has been considered as a case study. The MV side of the substation has been considered as a slack bus, with the voltage amplitude fixed to 1.0 p.u., and an angle equal to 0°. This assumption is coherent with the fact that an ideal OLTC has been considered in the PS, capable of constantly keeping the voltage to the reference value. The limits and constraints of the OLTC technology have not been implemented in the proposed algorithm.

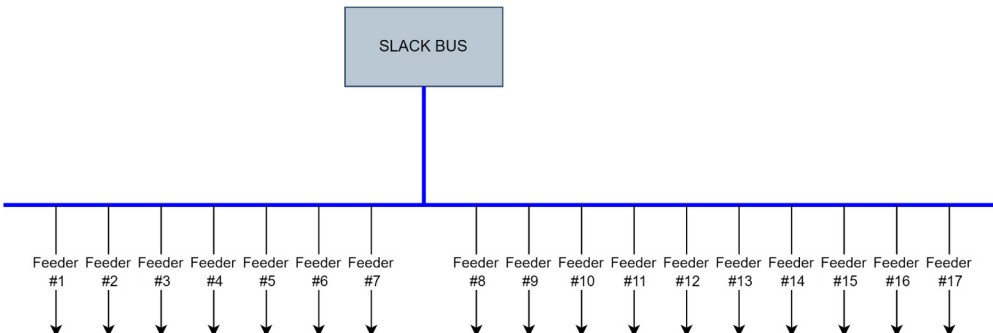

**Figure 14.** Schematic representation of the Aosta distribution system.

To test the effectiveness of the proposed algorithm, 24 h simulations with 1 h intervals were made by choosing a typical day for each season of the year, as shown in Figure 15 and reported in Table 2.

**Table 2.** Summary of the typical days chosen to test the algorithm proposed in the study case.

| Day | Consumption [MWh] | DG Production [MWh] | Reference Figure |
|---|---|---|---|
| 7 February | 562.755 | 191.890 | Figure 16 |
| 25 May | 355.964 | 132.565 | Figure 17 |
| 2 August | 407.274 | 44.732 | Figure 18 |
| 17 October | 472.981 | 196.267 | Figure 19 |

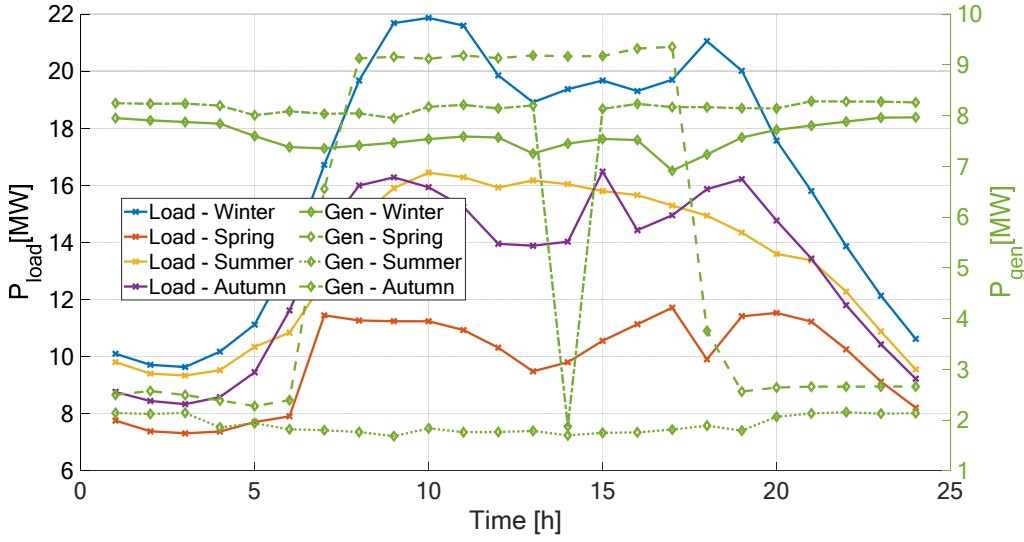

**Figure 15.** Cumulative real power loads absorption and generators production for each typical day.

The DN's production and hourly generation profiles come from data gathered on the field by the DSO. Power profiles were available for medium-voltage consumers, producers, and the PS. Data concerning the hourly consumption of secondary substations was not available. Equation (7) estimates the consumption of each secondary substation:

$$P_{t,i}^{ss} = \frac{P_t^{PS} - P_t^{MV}}{\sum_j S_j^{TRANS}} S_i^{TRANS} \tag{7}$$

where:

- $P_{t,i}^{SS}$ is the hourly power consumption of the *i*-th secondary substation (SS);
- $P_t^{PS}$ is the hourly power consumption of the primary substation (PS);
- $P_t^{MV}$ is the summation of the MV users' hourly consumption. This parameter also includes the distributed generation;
- $S_j^{TRANS}$ is the apparent power of the transformer installed in the *j*-th secondary substation.

The *m* penalty factor *C* has been set equal to $10^{-5}$. Results are summarised for the four seasons in Figure 16 (winter), Figure 17 (spring), Figure 18 (summer), and Figure 19 (autumn), where the MV nodes' voltage are plotted for each hour of the day, with respect to the equivalent electrical distance between the slack bus and the node, defined as the absolute value of the Thevenin equivalent impedance. In general, the proposed regulation provides a significant voltage improvement in all cases analysed, both hourly and seasonally; results also summarised in Table 3 in terms of Root Mean Square Error (RMSE).

**Table 3.** RMSE for nodal voltages with and without regulation.

| | | Voltages Root Mean Square Error | |
|---|---|---|---|
| | | **Without Regulation** | **With Regulation** |
| | Winter | 0.0036 | 0.0034 |
| Season | Spring | 0.0113 | 0.0084 |
| | Summer | 0.0087 | 0.0066 |
| | Autumn | 0.0052 | 0.0041 |

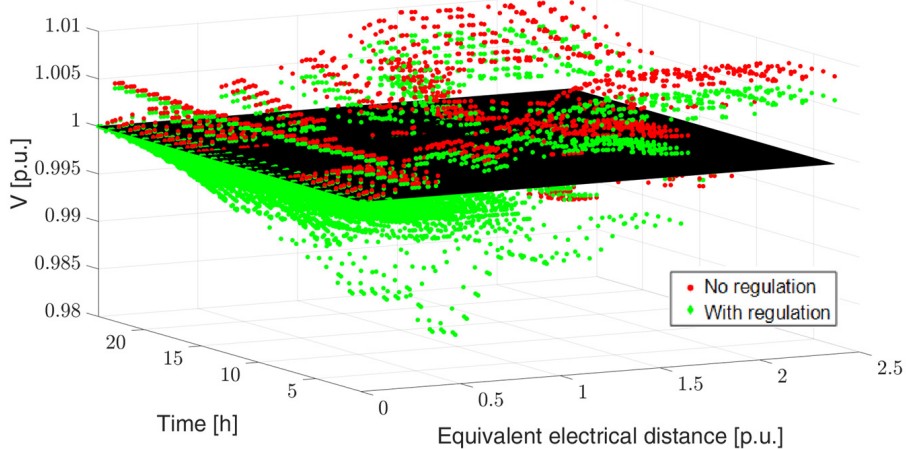

**Figure 16.** Impact of the voltage regulation proposed in winter.

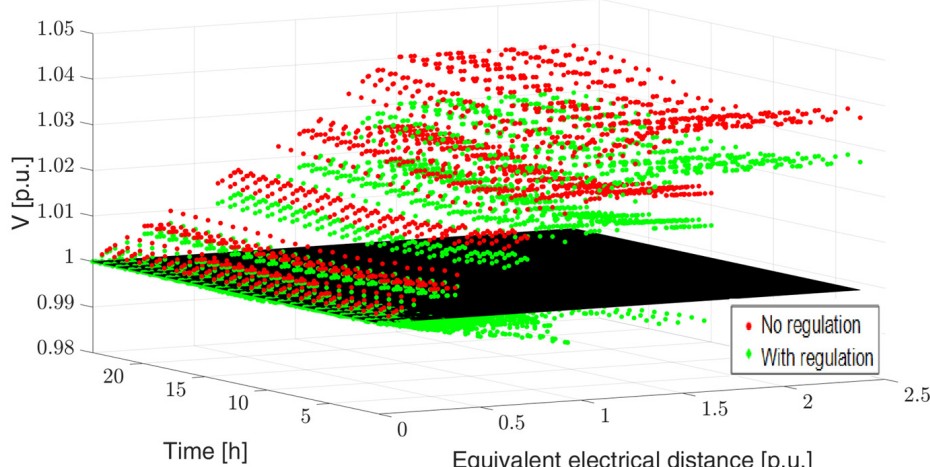

**Figure 17.** Impact of the voltage regulation proposed in spring.

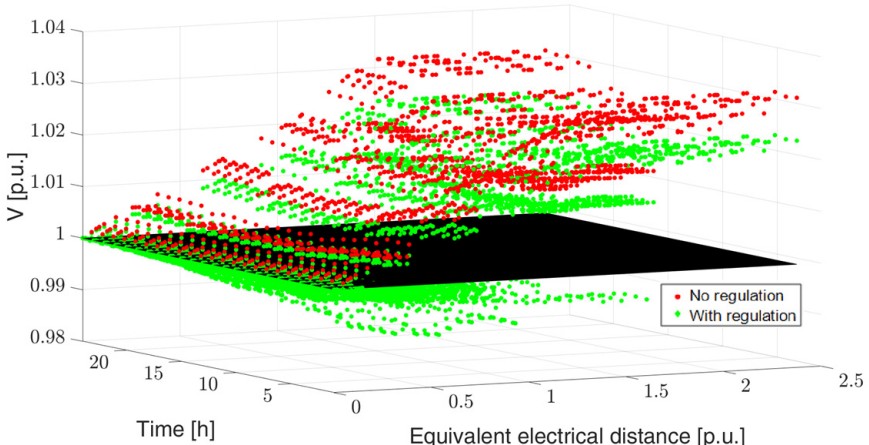

**Figure 18.** Impact of the voltage regulation proposed in summer.

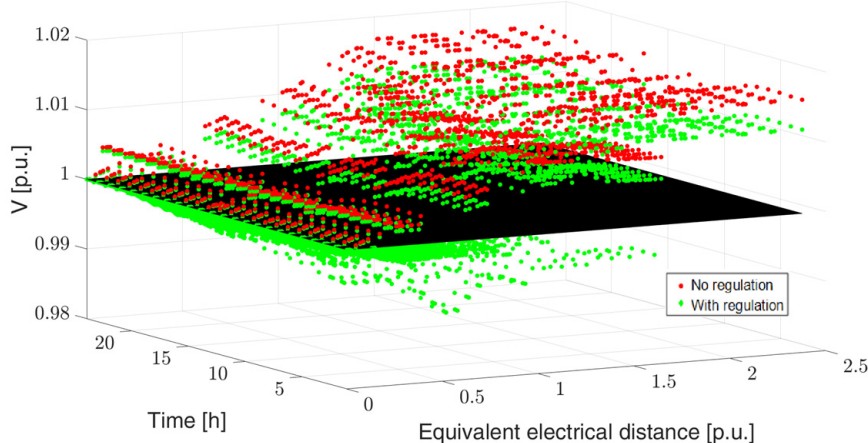

**Figure 19.** Impact of the voltage regulation proposed in autumn.

Lastly, the proposed method has been compared with a traditional fully centralised ORPF with the same objective function (i.e., the minimisation of voltage deviation). Numerical results are shown in Figure 20 for the winter day (for the sake of simplicity, other days are omitted). From the boxplots, it can be seen that, with the proposed decentralised approach, voltage profiles are little worse than the ones obtained with the centralised strategy: this is an expected result, since without (1), the reactive power withdrawn by

the generators is left free to change to minimise voltage deviations. However, voltage trends in the two cases are very close. Then, the advantages of the decentralised control in terms of adaptation to disturbances and lower data exchange requirements can motivate its adoption over the traditional one.

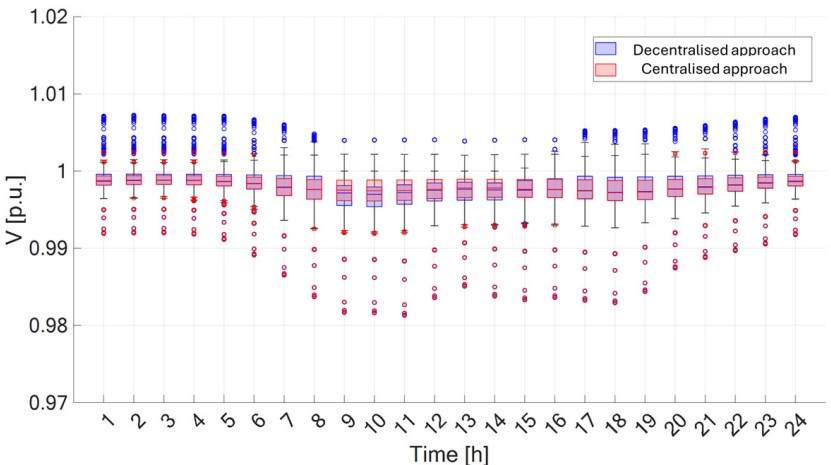

**Figure 20.** Comparison between the proposed method and a classical centralised ORPF for the winter day.

## 5. Conclusions and Future Studies

This research introduced a decentralised voltage regulation strategy tailored to address the challenges posed by the escalating integration of DERs, particularly inverter-based ones, driven by the remarkable surge in global renewable electricity generation, primarily through distributed PV.

The proposed strategy investigated a decentralised regulation by DERs to minimise voltage deviations across medium-voltage nodes; the implemented control law is aligned with the latest European technical standards and grid codes to best use DERs on the DNs.

The novelties proposed by the methodology reside in an ORPF capable of identifying the most efficient angular coefficient for each DER's Q(V) control law. The efficacy of the optimisation strategy was confirmed through several numerical simulations on both a small-scale network and a real Italian DN. The outcomes affirm the effectiveness of the decentralised approach in enhancing voltage profiles across diverse scenarios. In this regard, the work assessed that the Q(V) control law was influenced by the electrical distance to the PS and the DN's active power ratio (generated/absorbed). In addition, the proposed methodology was daily-tested in the four seasons: in all of them, the algorithm allowed a reduction in the voltage deviation at the MV nodes in a real-life DN. Furthermore, a comparison with a traditional ORPF was performed; despite the latter providing slightly better performance, the decentralised approach has clear benefits, concerning its capability to manage disturbances (eg., sudden voltage variations) without a real-time communication between the DSO and DERs, and the lower data exchange requirement.

In future works, the decentralised strategy will also be tested with other objective functions to compare the algorithm's outcomes and performance. In addition, other regulation resources could also be considered, such as the OLTC in the PS. Furthermore, different optimisation or metaheuristic algorithms could be compared or coupled to identify the optimal trade-off between accuracy and computational effort. Finally, investigating the integration of advanced communication and control mechanisms could further enhance the robustness of the decentralised voltage regulation strategy in dynamic grid environments.

**Author Contributions:** Conceptualisation, E.D., R.N., M.S., D.F., V.I. and M.M.; methodology, R.N. and M.S.; validation, D.F., V.I. and M.M.; resources, M.S. and M.M.; data curation, E.D., M.S. and R.N.; writing—original draft preparation, E.D., R.N., M.S., D.F., V.I. and M.M.; writing—review and editing, E.D., R.N., M.S., D.F., V.I. and M.M.; visualisation, E.D. and R.N.; supervision, D.F., V.I. and M.M.; project administration, D.F., V.I. and M.M. All authors have read and agreed to the published version of the manuscript.

**Funding:** This research received no external funding.

**Data Availability Statement:** Data are contained within the article.

**Acknowledgments:** The authors thank DEVAL S.p.A. for generously providing access to the necessary information and technical knowledge to support this research.

**Conflicts of Interest:** The authors declare no conflicts of interest.

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
