# Peer review of "Decentralised Voltage Regulation through Optimal Reactive Power Flow in Distribution Networks with Dispersed Generation"

_electricity, doi:10.3390/electricity5010008_

Round 1
Reviewer 1 Report
Comments and Suggestions for Authors
This article contains analyzes of voltage regulation in RES-saturated power grids. The authors proposed their own optimal methodology to improve voltage profiles. This type of analysis is important because it improves the process of managing the distribution network in terms of voltage regulation.
• The structure of the manuscript is correct: 1) Introduction to the research topic; 2) Literature review; 3) Methodology; 4) Calculation cases and results; 5) conclusions and future research. The abstract is formulated correctly and contains a short description of the research topic. The keywords are selected appropriately.
• The literature review is correct. I believe that the number of works cited is appropriate and sufficient. The authors analyzed these articles from the point of view of the research topic under consideration. They divided the cited works into those related to voltage regulation and optimization.
However, there are several issues that should be considered and clarified:
• The research is correctly designed, but the authors should consider more computational cases. They should also analyze the different operating states of the distribution network:
o In a real power grid there are extreme states, i.e. low load and high generation. In such cases, voltage regulation only using reactive power may not be sufficient. Does the proposed algorithm also take into account the possibilities of controlling the active power of DER? There are networks that are lightly loaded, with a large number of DERs and long sections of lines with small cross-sectional cables. If the simultaneous generation of a large number of DERs is high, it may turn out that regulating only reactive power will not be sufficient.
o Why is Vref = 1 p.u.? In my opinion, Vref should be adjusted to real conditions (usually voltage values are higher).
o Table number 1: Why is the load power 7MW? Is this due to the load capacity of the line?
o The Aosta Case Study: In my opinion, the authors should make charts for total generation and total load for the considered time periods (hourly charts).
o The Aosta Case Study: In my opinion, the authors should make graphs of voltage variations in network nodes before optimization (on one graph of voltage changes in all network nodes).
• In my opinion, the research methodology should be described in more detail, e.g.:
o Detailed description of the proposed methodology, e.g. block diagram,
o A more detailed description of how to determine the m factor,
o Is the proposed method universal? Will it be effective in the case of networks with a complex structure and a large number of branches?
o Were the penalty coefficients C determined arbitrarily?
• The font color of the text in the captions of figures and tables is different from the font color of the main text - in my opinion, the font color should be black. In the current version, the font color (contrast) is different than the font color of the main text.
• Figure no. 8, figure no. 9, figure no. 10, figure no. 11 – too small font size of the text on the charts (small font next to the axis caption, small font next to the numbers).
• Some lines have single letters at the end, such as "a" or the articles "the". They should appear at the beginning of the next line.
• The authors use the symbol "*" as the multiplication symbol (e.g. equation (1), table number 1 (first row and second column of the table). In my opinion, it would be better to use the symbol "·" as the multiplication symbol.
• The presented conclusions are consistent with the assumptions. The authors actually confirm that they have achieved their goal, i.e. proving the effectiveness of the proposed algorithm. There are no such methods in practice, especially in medium- and low-voltage distribution networks. This is particularly important in the case of power grids saturated with RES. With such a tool, you will be able to control tension problems more effectively. Based on the presented results, it can be concluded that the proposed methodology is effective. However, the summary should also clearly indicate what new information the article brings compared to other works of this type. Why is the proposed methodology original and worth further analysis? In this version, point 4 is mostly a description of what was done in the article. Additionally, possibilities to improve the proposed methodology in future work are described.
Reviewer 2 Report
Comments and Suggestions for Authors
The authors present A Decentralized Voltage Regulation through ORPF in Distribution Networks with Dispersed Generation, which is very interesting. However, the following comments should be met.
1. Based on a reference found which is http://dx.doi.org/10.1016/j.eswa.2017.06.009, the ORPF problem aims to minimize both voltage deviation and line losses, and also based on the references provided in section 2.2, many papers considered several objectives. However, in the paper, the authors only aim to minimize the voltage deviation. Please explain why the authors only consider one objective and is it better to include line losses as the objective functions.
2. The proposed method has been tested to solve the problem; however, there is no any compared algorithm. How the authors verify the proposed method if it generated results are correct or not. Some other algorithms should be used to compared the results, or the results can be compared with other literature.
3. The main contributions of this work should be highlighted in the introduction section.
4. Some significant numerical results should be added to the abstract.
5. The abbreviation DERs has not been defined in the Abstract. All abbreviations must be rechecked through out the paper.
6. English grammar and typo must be carefully rechecked.
Comments on the Quality of English LanguageModerate editing of English language required
Reviewer 3 Report
Comments and Suggestions for Authors
The manuscript is well organized and has detailed information and presents a solution to the voltage drop problem in distributed energy management under Italian energy regulations and grid. The authors proposed a customized Optimal Reactive Power Flow (ORPF) that, optimally setting up the droop coefficient of each DER, allows minimizing the deviation between the voltage value in each node and its nominal value, whilst taking advantage of the specific control capabilities of each generator. They tested tested on both a small-scale test grid and on a real Italian distribution grid to propose an efficient solution to the voltage regulation problem. The reviewer thanks to authors for their comprehensive and well-written paper.
There are minor concerns that must be reconsidered to improve the content of a paper comprehensively. Nevertheless, the reviewer hopes the authors will find the below-given comments helpful to enhance their study:
· Since the lines in the graphs from Figure 4 to Figure 7 in Section 4.1.1 are very thin, their clarity is low. It would be appropriate to thicken them.
· The results for four seasons should explained under the relevant Figures. (Figures 14-17)
Comments on the Quality of English LanguageThe English of the paper is good, but a native speaker should recheck it.
Round 2
Reviewer 1 Report
Comments and Suggestions for Authors
Thank you for your replies and constructive corrections. The article is very good. Good luck.
Reviewer 2 Report
Comments and Suggestions for Authors
The authors have responsed all of my comments. I have no more comment.
Comments on the Quality of English LanguageMinor editing of English language required